# Health information technology uses for primary prevention in preventive medicine: a scoping review protocol

Abrar Alturkistani,[1] Azeem Majeed,[2] Josip Car,[1] David Brindley,[3] Glenn Wells,[4] Edward Meinert[1,3]

¹Global Digital Health Unit, Department of Primary Care and Public Health, Imperial College London, London, UK
²Department of Primary Care and Public Health, Imperial College London, London, UK
³Healthcare Translation Research Group, Department of Paediatrics, University of Oxford, Oxford, UK
⁴Oxford Academic Health Science Centre, Oxford, UK

**Correspondence to**
Edward Meinert;
e.meinert14@imperial.ac.uk;
edward.meinert@paediatrics.ox.ac.uk

## ABSTRACT

**Introduction** The use of health information technologies (HITs) has been associated with positive benefits such as improved health outcomes and improved health services. Results from empirical studies reported potential benefits of HITs in preventive medicine measures such as primary prevention. This review will examine the broad range of HITs and their uses and effectiveness in primary prevention.

**Methods and analysis** We will conduct searches in relevant databases (MEDLINE, EMBASE, the Cochrane Methodology Register, Cochrane Database of Systematic Reviews, CINAHL, SCOPUS and Web of Science) using Arksey and O'Malley's scoping review methodology. The scoping review will include all study designs to identify the literature on HIT uses. Two reviewers will independently screen the literature following our screening criteria and using a data abstraction form. Findings will be summarised quantitatively (using numerical counts of HITs) and qualitatively (using narrative synthesis).

**Ethics and dissemination** The study will synthesise data from published literature and will not require an ethical approval. The results of the review will be disseminated through a peer-reviewed journal.

## BACKGROUND

Health information technology (HIT) includes technologies that enable health information to be stored, disseminated and analysed[1] and are increasingly used to improve the health of patients and populations. Popular examples of HITs include electronic health records, smartphone health applications (apps) and electronic prescriptions (E-prescribing).[1] Evidence from existing systematic reviews and empirical studies found positive effects of using HITs in improving health outcomes. Research shows that HITs can not only improve health outcomes but also contribute to preventing disease and improving preventive medicine practices. Preventive medicine is the practice that focuses on keeping individuals healthy, and its goal is to 'protect, promote, and maintain health and well-being and

### Strengths and limitations of this study

- ► This study will conduct a comprehensive review of the relevant databases to help inform healthcare professionals, researchers and policy-makers about the latest uses of health information technologies (HITs) for preventive medicine purposes.
- ► It will also help identify gaps in the literature concerning HITs and their effectiveness and uses in preventive medicine.
- ► It will only include English language publications.
- ► It will not perform a formal quality assessment of included studies.

to prevent disease, disability, and death'.[2] Primary prevention is one of the preventive medicine measures, and it is defined as the prevention of 'the initial occurrence of a disorder' by the WHO.[3] Despite the potential benefits that HITs can have to improve primary prevention, and the availability of studies about the use of HITs for primary prevention, there are currently no studies that comprehensively review the different types of HITs and their uses in primary prevention.

HITs have seen a growing interest in the literature in recent years and have been repeatedly associated with preventing disease[4–6] improving health outcomes,[7] improving data collection and the potential to substantially advance healthcare research.[8–10] As different HITs proliferate, questions about their effectiveness are being raised. HITs are associated with positive outcomes in healthcare in general such as 'efficiency of care', 'effectiveness of care' and 'patient safety'.[10]

Reviews related to the use of HITs in primary prevention focus on only one or two types of HITs (eg, telephone-based interventions only).[11] Most of the studies that focus on primary prevention outcomes focus on one tool or method of HITs like

**Table 1** Description of preliminary list of existing health information technology uses in primary prevention

| Intervention | Primary prevention uses | Description of intervention |
|---|---|---|
| Mobile phone messaging (SMS or MMS) | Smoking cessation Rodgers et al[9] | Personalised smoking-related and general healthy behaviour-related messages sent to participants as part of a smoking cessation programme. The intervention had other features like being able to text other participants, requesting texts on quitting-related tips and taking polls and quizzes about smoking.[9] |
| | Adherence in taking vitamin C for preventive reasons Cocosila et al[13] | Text message sent from a virtual character to remind to take a vitamin C pill to participants, where they were expected to 'acknowledge' the reminder. If the text was acknowledged, an encouraging message is sent, if not, a reminder message is sent. The encouraging messages were described as amusing while the reminder messages were described as 'non-amusing'.[13] |
| | Healthy behaviour in children Shapiro et al[14] | Feedback text messages sent as part of a programme to promote healthy behaviours in children (to increase physical activity, reduce sugary beverage consumption and screen time). The feedback text messages were sent once the participants sent a text message informing their achievement of predetermined healthy behaviour-related goals.[14] |
| Internet-based interventions | Smoking prevention Buller et al[15] | Internet-based programme for school-children that uses 'audio narration, graphics, animation, sound effects, and music' to deliver lessons for smoking prevention with survey questions asked to personalise the lessons for the student.[15] |
| | HIV prevention Kasatpibal et al[16] | Internet-based educational programme that uses 'texts, pictures, animation, animated cartoons, videos, message boards, and exercise' to teach about the risks of HIV for men who have sex with men.[16] |
| | Obesity prevention Rerksuppaphol and Rerksuppaphol[17] | Internet-based programme for school-aged children to track weight and nutrition-related information and provide personalised information about nutrition and physical activity based on the user's weight/health status.[17] |
| Telephone-based intervention | Postpartum depression prevention Lewis et al[18] | A telephone-based intervention to increase exercise (known to prevent postpartum depression) as part of a prevention programme. The telephone-based intervention is used to inform and educate the participants about exercising, explain exercise recommendations and encourage participants to maintain exercising.[18] |
| Smartphone application (app) | Diabetes prevention Fukuoka et al[19] | An interactive app with a 'self-monitoring' tool and a list of tasks for activities that can prevent diabetes like physical activity. The app also provides encouraging feedback based on the user's input.[19] |

MMS, multimedia messaging service; SMS, short message service.

electronic health records[8] or mobile health technologies.[5] However, these studies are not representative of the whole range of HITs that can be used in primary prevention. In addition, some of the currently available reviews, even if include more than one HIT, only focus on one or two primary prevention outcomes (eg, smoking).[9]

This review will focus on gathering information on what is available rather than which interventions work best. This general focus allows the examination of all the available interventions in HITs. In this review, we will map out the findings and results of studies published about HITs and their uses in primary prevention in preventive medicine. A scoping review can help clarify to what extent are HITs used for primary prevention purposes, and what is the range of the HITs available. We will synthesise the available evidence to inform how technology could be developed to impact primary prevention in preventive medicine. In this protocol, we have reviewed some HITs used for primary prevention in table 1, as examples of the scoping review outcomes that will result from the study.

## AIMS AND OBJECTIVES

The aim of this review is to provide an overview of all HITs that are used for the purpose of primary prevention or to achieve primary prevention outcomes. Through this review, the available HITs, their uses, limitations and gaps in the literature regarding their use in primary prevention will be reported. The objectives of the review are the following:

▶ To identify the HITs that are used for primary prevention and to analyse both the benefits and risks achieved by their use.
▶ To identify the primary prevention patient outcomes that are impacted by the use of HITs.

## METHODS

To outline the protocol of the forthcoming scoping review, we will be using the Preferred Reporting Items for Systematic Reviews and Meta-Analyses for Protocols (online supplementary appendix 1).

### Protocol design

We will use the Arksey and O'Malley methodological framework for scoping reviews in performing the

 Alturkistani A, et al. BMJ Open 2018;8:e023428. doi:10.1136/bmjopen-2018-023428

| Table 2 | Scoping review primary and secondary research questions |
|---|---|
| **Primary research questions** | **Secondary research questions** |
| What HITs are used in primary prevention in preventive medicine to impact individuals/patients health outcomes? | ► What tools and innovations of HITs are used in primary prevention in preventive medicine?<br>► What primary prevention in preventive medicine patient/individual health outcomes are impacted by the use of HITs?<br>► What are the risks and benefits associated with HITs?<br>► How are the use of HITs changing/improving primary prevention in preventive medicine compared with standard/traditional methods? |

HITs, health information technologies.

review. The framework recommends the following six steps to conduct a scoping review: (1) identifying the research question; (2) identifying relevant studies; (3) selecting studies; (4) charting the data; and (5) collating, summarising and reporting the results.[12] This framework is being used for this review because it applies a rapid form of knowledge synthesis, with the intent to identify the merits of the underlying research question. This form of review is intended to be a precursor for potential further work, as on initial analysis it is unclear if a more sophisticated review method is warranted.

### Stage 1: identifying the research question
The preliminary research (table 1) revealed that there are no review studies that reviewed the different HIT approaches used in primary prevention and exposed a research gap that motivated the focus of this protocol. The main research question and the secondary research questions of the scoping review are displayed in table 2.

### Stage 2: identifying relevant studies
*Search strategy*
We will conduct searches in relevant electronic databases: MEDLINE, EMBASE, the Cochrane Methodology Register, Cochrane Database of Systematic Reviews, CINAHL, SCOPUS and Web of Science. The initial literature search strategy used for MEDLINE can be found in online supplementary appendix 2, including the medical subheadings (MeSH) and free-text terms used to perform the search. The search strategy will be modified for each database and further iterated as we explore the research question with changes captured in the review process. Studies will not be limited in terms of year or study design. Only studies in English language will be reviewed. Apart from electronic databases, we will also search reference lists of the studies selected for full-text reading to supplement the search.

### Stage 3: Study selection
Screening of the studies will be performed by two suitably experienced/qualified reviewers and in two levels. Table 3 outlines the inclusion criteria that will be used by the reviewers to determine the studies that will be included. The citation management software program, EndNote X8.2 (Clarivate Analytics, USA), will be used to manage records and data and to remove duplicates. The first screening will involve screening the title and abstracts. Using two reviewers will ensure that all relevant articles are included. The reviewers will use the predefined relevance criteria to determine relevant studies. In the second round of screening, the reviewers will perform full-text reading of the studies identified in the previous round. Conflicts and discrepancies will be resolved by discussing with a third party.

| Table 3 | Review inclusion criteria |
|---|---|
| **Inclusion criteria** | |
| Population | ► Users of the health information technologies will include individuals or patients who are treated with primary prevention in preventive medicine. |
| Intervention | ► All health information technologies (eg, electronic health records, telemedicine, text messages, computerised decision support systems). |
| Comparator | ► Studies using non-health information technology interventions.<br>► Studies using traditional or usual method as a comparator to health information technology.<br>► Studies without a comparator. |
| Outcomes | ► Any primary prevention outcome that prevents a disease or a health-threatening condition or a behaviour before it occurs (eg, chronic disease prevention, smoking prevention, obesity prevention). |
| Study type | ► Any study type; experimental (randomised controlled trials (RCTs), quasi-RCTs, non-RCTs), quasi-experimental (controlled before–after, interrupted time series) and observational (cohort, case–control, cross-sectional) and review (systematic review, meta-analysis scoping review) studies.<br>► Only publications in English will be included.<br>► There will be no restrictions to calendar date; we intend to capture a broad survey of technologies developed and therefore are not restricting date range. |

**Table 4** Data analysis plan by the synthesis objectives and anticipated outputs

| Synthesis objective | Method | Guide questions | Outputs |
|---|---|---|---|
| 1. To identify the health information technologies that are used for primary prevention. | We will summarise the identified studies by the health information technology used. | What is the health information technology? What is the purpose of the health information technology and how does the purpose contribute to primary prevention? In what setting is the primary prevention technology used? (Eg, healthcare, community setting…, etc.) What type of evidence does the study provide for primary prevention-related health outcomes? | A list of the health information technologies used for primary prevention purposes. A list of the settings that the health information technologies are used in categorisation of the primary prevention-related outcomes. |
| 2. To identify the primary prevention patient outcomes that are improved by the use of health information technologies. | We will strictly identify the studies that reported significant improved patient outcomes as a result of using health information technologies. | What are the studies that reported significant improved patient outcomes and what is the criteria they used to represent significance? How health information technologies that improve patient outcomes are used to improve primary prevention measures? Are there any disadvantages of using the health information technologies for primary prevention? Can the health information technology be translated and used in different healthcare-related settings? | Identification of the health information technologies that contribute significant improved patient outcomes in the literature. A thematic report of the health information technology uses in primary prevention. |
| 3. Map out the ways health information technologies are changing/improving primary prevention compared with standard/traditional methods. | We will identify the articles that compare health information technology interventions to traditional or standard interventions. | Did the study compare primary prevention health outcomes to other standard or traditional methods of primary prevention? What outcomes did the study report to compare the health information technologies to other methods? How long were the health information technologies and other methods compared for? | A summary of the health information technologies that were reported to have superior primary prevention outcomes when compared with traditional or standard methods to map out the specific health information technologies that have been compared with traditional or standard methods of primary prevention. |

*Exclusion criteria*
► Interventions that focus on secondary or tertiary prevention will be excluded to keep the focus on the primary prevention interventions only.
► Publications that are not in English will be excluded.

## Stage 4: charting the data

Two reviewers will independently extract the data and vigilantly review the studies based on the data abstraction form (online supplementary appendix 3). We assume that studies identified for this review will include basic study information like: first author and year of publication and will include information about the HIT intervention and the methods used in the study. Following review of the primary study types to be included in the review, an appropriate quality-assessment standard shall be used to assess the quality of the included papers.

## Stage 5: collating, summarising and reporting the results

The studies identified from this scoping review will be summarised and analysed using quantitative and qualitative methods. In terms of quantitative methods, we will report simple numerical counts of information such as: the total number of studies, types of primary prevention HIT interventions, descriptions of the study samples and regarding qualitative methods, we will conduct a narrative synthesis to provide an overview of the breadth of the literature and to identify gaps that may need further research. To address the three research questions of the review, we will analyse the data following three synthesis objectives: to identify the HITs that are used for primary prevention, to identify the primary prevention patient outcomes that are improved by the use of HITs and to map out the ways HITs are changing/improving primary prevention compared with standard/traditional methods. Table 4 displays each of the synthesis objectives of the review followed by the method, guide questions and outputs that will be used to achieve them.

## Patient and public involvement

Research interests identified and prioritised by members of the public in a workshop held at the European Scientific

Institute in July 2017 were used to guide specification of this research.

## Ethics and dissemination

The proposed scoping review has the potential to improve research and inform policy-makers, healthcare providers, clinicians and researchers on how HITs are used in preventive medicine. This scoping review could help advance research by showing the type of evidence and strategies available and by highlighting the need for further research in the field. The completed scoping review will be disseminated via publication in a peer-reviewed journal to categorise HITs for their use in primary prevention.

**Acknowledgements** We thank the medical librarians at Imperial College, Charing Cross campus for advising on search strategies and available resources. This work was supported by the Sir David Cooksey Fellowship in Healthcare Translation and the SENS Research Foundation.

**Contributors** AA and EM participated in the design and development of the protocol. AA and EM drafted the manuscript. AM, JC, DB and GW reviewed the second draft. AA and EM incorporated and addressed the feedback from the authors. All authors read and approved the final manuscript.

**Funding** This work was funded by EIT Health (Grant 18654).

**Competing interests** None declared.

**Patient consent** Not required.

**Ethics approval** Due to the use of publicly available, published data, this study will not require an ethical approval.

**Provenance and peer review** Not commissioned; externally peer reviewed.

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
