## [Reviewer comments · BMJ Open]

ARTICLE DETAILS

TITLE (PROVISIONAL)	Health information technology uses for primary prevention in preventive medicine: A scoping review protocol
AUTHORS	Alturkistani, Abrar; Majeed, Azeem; Car, Josip; Brindley, David; Wells, Glenn; Meinert, Edward

VERSION 1 – REVIEW

REVIEWER	Karen Day The University of Auckland, New Zealand
REVIEW RETURNED	08-May-2018

GENERAL COMMENTS	Thank you for the opportunity to review this scoping literature review protocol. The proposed literature review's aim is to scope the health information technologies (HIT) that improve health outcomes as a result of primary preventive features and intentions. Overall, this is an interesting proposal and worth conducting. I do, however, have some concerns. The objectives are to examine outcomes that are improved by the use of the technologies. To my mind it is better to examine benefits and risks together. Consequently, in the discussion section of your report you will be able to weigh the benefits and risks to determine how outcomes are improved or not by specific HIT interventions. I argue that the scope of your review is too narrow and is at risk of becoming part of the phenomenon of 'publication bias' (Easterbrook, Gopalan, Berlin, & Matthews, 1991; Rothstein, Sutton, & Borenstein, 2006) or at the very least contributing to publication bias. Your choice of the Arksey and O'Malley methodological framework does not have a rationale other than to say that it's designed for scoping reviews. In light of your review (exploring relatively uncharted waters in HIT research on preventive technologies), I recommend that you at least provide some evidence in your protocol that you have examined the potential relevance of other literature review methods. For example, systematic reviews are considered the gold standard in medical research and should be considered before being discarded and a rationale should be supplied. Furthermore, the meta-narrative review process by Greenhalgh et al (2009) provides ways to uncover unexpected results, synthesise disparate reports on similar technologies, and explore emerging technologies. It may not serve your needs, but a reason for not using it could strengthen your choice to use the Arksey and O'Malley methodological framework. Pare et al (2015) provide a good typology of literature reviews that includes a critique of the Arksey and O'Malley approach that could be useful. You say that quality assessment of the literature will not be done because this is not what a scoping review includes. According to Pare et al (2015) this
---

	is not necessarily so, and that the Arksey and O'Malley approach is criticised for not enabling a quality assessment. Since you are going to search through the Cochrane database, why are you not using the Cochrane process? You may in the end still settle for the Arksey and O'Malley approach but it will be an informed decision. It is good to see the use of the PRISMA checklist, and the specific search strings developed for the databases you will be using. Since preventive medicine is not limited to doctors, why are you not including CINAHL in your database search? You're not likely to find much because the emphasis in HIT development is on medical (doctor) clinicians, but you could find some literature that might be worth including. Since the technologies themselves are often developed outside of clinical care, it might be useful to at least indicate an exploratory search in the business and computer science literature databases to indicate that you have done due diligence, e.g. Business Source Premier (there is an entire discipline of 'information systems' in the business field and often their literature is applicable in health and is overlooked). SCOPUS and Web of Science may deliver results that you may not find in Medline or EMBASE – it is rare that this happens but you may find a gem. In table 3 you say that 'only published literature' will be used, which is ambiguous and could include grey literature. Do you mean 'peer reviewed published literature'? Inclusion criteria – do you have a date range for including literature and if so why that date range, or no date range? Medline search strategy and terms: As a preliminary search query, this appears to be comprehensive enough for your scoping study. I expect it would need to be tweaked as you find literature and unexpected language that will need to be included. Please add a statement to this effect. One expects things to change a little as you progress through your planned research, and adding a statement to this effect signals a sensible approach. All the best with your literature review! It looks very interesting and worthwhile. References Easterbrook, P. J., Gopalan, R., Berlin, J., & Matthews, D. R. (1991). Publication bias in clinical research. The Lancet, 337(8746), 867-872. Greenhalgh, T., Potts, H. W. W., Wong, G., Bark, P., & Swinglehurst, D. (2009). Tensions and paradoxes in electronic patient record research: A systematic literature review using the meta-narrative method. The Milbank Quarterly, 87(4), 729 - 788. Paré, G., Trudel, M.-C., Jaana, M., & Kitsiou, S. (2015). Synthesizing information systems knowledge: A typology of literature reviews. Information & Management, 52(2), 183-199. Rothstein, H. R., Sutton, A. J., & Borenstein, M. (2006). Publication bias in meta-analysis: Prevention, assessment and adjustments: John Wiley & Sons.
--	--

VERSION 1 – AUTHOR RESPONSE

Reviewer Feedback	Author response	Action taken
Thank you for the opportunity to review this scoping literature review protocol. The proposed literature review’s aim is to scope the health information technologies (HIT) that improve health outcomes as a result of primary preventive features and intentions. Overall, this is an interesting proposal and worth conducting. I do, however, have some concerns.	We thank the reviewer for her time and consideration of our manuscript. Her review comments are well reasoned and have resulted in changes to the paper which will strengthen the study.	N/A
The objectives are to examine outcomes that are improved by the use of the technologies. To my mind it is better to examine benefits and risks together. Consequently, in the discussion section of your report you will be able to weigh the benefits and risks to determine how outcomes are improved or not by specific HIT interventions. I argue that the scope of your review is too narrow and is at risk of becoming part of the phenomenon of ‘publication bias’ (Easterbrook, Gopalan, Berlin, & Matthews, 1991; Rothstein, Sutton, & Borenstein, 2006) or at the very least contributing to publication bias.	The reviewer makes an important point; in order to control technological determinism and a direct implication that technology always has a positive outcome, the language in this section has been modified to analyse both strengths and weaknesses and the impact of these technologies. The reviewers point regarding publication bias is understood and intent noted. The reason for the narrow focus for this review is because of the authors specific work in digital health and focus on better understanding the design and development of digital solutions in this content. In order to control bias, the RQs and aims and objectives have been reworded to reflect a careful examination of strengths and weaknesses. Such an approach will not lead to a default laudatory review of technology.	 1. Removed an objective which could bias results in favour of technology 2. Revised aims and objectives to reflect both risk and benefits and modified language to analyse impact versus direct benefit 3. The research questions have been modified to review impact and analyse risk/benefit.

Your choice of the Arksey and O'Malley methodological framework does not have a rationale other than to say that it's designed for scoping reviews. In light of your review (exploring relatively uncharted waters in HIT research on preventive technologies), I recommend that you at least provide some evidence in your protocol that you have examined the potential relevance of other literature review methods. For example, systematic reviews are considered the gold standard in medical research and should be considered before being discarded and a rationale should be supplied. Furthermore, the meta-narrative review process by Greenhalgh et al (2009) provides ways to uncover unexpected results, synthesise disparate reports on similar technologies, and explore emerging technologies. It may not serve your needs, but a reason for not using it could strengthen your choice to use the Arksey and O'Malley methodological framework. Pare et al (2015) provide a good typology of literature reviews that includes a critique of the Arksey and O'Malley approach that could be useful. You say that quality assessment of the literature will not be done because this is not what a scoping review includes. According to Pare et al (2015) this is not necessarily so, and that the Arksey and O'Malley approach is criticised for not enabling a quality assessment.	The reviewer raises a fundamental point concerning the importance of method selection in review analysis. The rationale for the Arksey and O'Malley framework is its open nature and objective to be a rapid form of knowledge synthesis. It is unclear at this stage if a more sophisticated review method is warranted and for this reason this method was selected. The narrative of the manuscript has been modified to reflect the reason for this framework selection. Additionally, quality appraisal will be completed following a review of the included studies and appropriate review standard will be utilised.	1. Detail added to protocol design justifying the selection of the scoping method 2. Update added to text to indicate selection of a quality appraisal following study selection
Since you are going to search through the Cochrane database, why are you not using the Cochrane process? You may in the end still settle for the Arksey and O'Malley approach but it will be an informed decision.	Same comment as previous review comment	N/A

It is good to see the use of the PRISMA checklist, and the specific search strings developed for the databases you will be using.	We thank the reviewer for this observation. The completed review will include a PRISMA flow diagram and the additional quality analysis amended in this revised manuscript.	N/A
Since preventive medicine is not limited to doctors, why are you not including CINAHL in your database search? You're not likely to find much because the emphasis in HIT development is on medical (doctor) clinicians, but you could find some literature that might be worth including. Since the technologies themselves are often developed outside of clinical care, it might be useful to at least indicate an exploratory search in the business and computer science literature databases to indicate that you have done due diligence, e.g. Business Source Premier (there is an entire discipline of 'information systems' in the business field and often their literature is applicable in health and is overlooked). SCOPUS and Web of Science may deliver results that you may not find in Medline or EMBASE – it is rare that this happens but you may find a gem.	The reviewer makes key points on databases to include. We have modified the search databases to broaden the search footprint following secondary consultation with the Imperial College Medical Librarian.	1. CINAHL, SCOPUS and Web of Science were added to the databases search strategy.
In table 3 you say that 'only published literature' will be used, which is ambiguous and could include grey literature. Do you mean 'peer reviewed published literature'?	The reviewer catches a drafting mistake. This sentence has been removed	1. The sentence referenced has been removed.
Inclusion criteria – do you have a date range for including literature and if so why that date range, or no date range?	We have modified the text to reflect the rationale for no date restriction; chiefly to capture a broad understanding of technologies used.	1. The sentence referenced has been modified to including additional detail for the lack of date restriction.

Medline search strategy and terms: As a preliminary search query, this appears to be comprehensive enough for your scoping study. I expect it would need to be tweaked as you find literature and unexpected language that will need to be included. Please add a statement to this effect. One expects things to change a little as you progress through your planned research, and adding a statement to this effect signals a sensible approach.	The reviewer makes a good suggestion to include iteration reflecting our initial search.	1. The referenced section has been modified to indicate iteration in search strategy in line with the protocol research questions.
--	---	---

VERSION 2 – REVIEW

REVIEWER	Karen Day The University of Auckland, New Zealand
REVIEW RETURNED	02-Aug-2018
GENERAL COMMENTS	Thank you for making the changes as recommended. I spotted a typo in the change to the 'Contributor statement'. Do you mean 'reviewer' instead of 'author' in the sentence "AA and EM incorporated and addressed the feedback from the authors"?